# Multi-Objective Optimization Applications in Chemical Process Engineering: Tutorial and Review

**Gade Pandu Rangaiah** [1,2,*] , **Zemin Feng** [1,3] **and Andrew F. Hoadley** [4]

1   Department of Chemical and Biomolecular Engineering, National University of Singapore, Singapore 117576, Singapore; fzm@cqu.edu.cn
2   School of Chemical Engineering, Vellore Institute of Technology, Vellore 632014, India
3   School of Chemistry and Chemical Engineering, and National-Municipal Joint Engineering Laboratory for Chemical Process Intensification and Reaction, Chongqing University, Chongqing 400044, China
4   Department of Chemical Engineering, Monash University, Clayton, Victoria 3800, Australia; andrew.hoadley@monash.edu
*   Correspondence: chegpr@nus.edu.sg or gprangaiah@gmail.com; Tel.: +65-6516-2187

**Abstract:** This tutorial and review of multi-objective optimization (MOO) gives a detailed explanation of the 5 steps to create, solve, and then select the optimum result. Unlike single-objective optimization, the fifth step of selection or ranking of solutions is often overlooked by the authors of papers dealing with MOO applications. It is necessary to undertake a multi-criteria analysis to choose the best solution. A review of the recent publications using MOO for chemical process engineering problems shows a doubling of publications between 2016 and 2019. MOO applications in the energy area have seen a steady increase of over 20% annually over the last 10 years. The three key methods for solving MOO problems are presented in detail, and an emerging area of surrogate-assisted MOO is also described. The objectives used in MOO trade off conflicting requirements of a chemical engineering problem; these include fundamental criteria such as reaction yield or selectivity; economics; energy requirements; environmental performance; and process control. Typical objective functions in these categories are described, selection/ranking techniques are outlined, and available software for MOO are listed. It is concluded that MOO is gaining popularity as an important tool and is having an increasing use and impact in chemical process engineering.

**Keywords:** multi-objective optimization; multiple criteria; Pareto optimal front; non-dominated solutions; chemical engineering; process engineering; optimization techniques; optimization procedure; optimization software; Pareto ranking

## 1. Introduction

Vilfredo Pareto, who is often referred to as the founder of socioeconomics, was also the first to publish on multi-objective optimization (MOO). He famously stated that in a closed society, "you could not make any individual wealthier without making another individual poorer", and he demonstrated this with the now famous Pareto plot [1]. This principle of monetary balance is used in chemical engineering (ChE), which is also known as chemical process engineering (CPE), in the balance of mass and energy for a defined system. It is the foundation of unit operations and, more widely, chemical process engineering. Despite the need to balance both mass and energy inputs, it was not until there was a need to optimize multiple outcomes that Pareto's MOO has risen to prominence and, notably in CPE, this is only in the last 20 years. This review paper will outline systematic steps for MOO, the development of MOO in CPE, and its use for technical process improvement and also for wider sustainability improvements, common techniques for MOO of processes, objectives used in process optimization, and available software for MOO.

A simple example would consider a factory making more than one product from a single raw material. It could be raw cotton for shirts or trousers, crude oil for gasoline or diesel, or distilled spirit (ethanol) as a fuel additive or alcoholic beverage such as whisky or gin. A single-objective optimization (SOO) approach would be to just direct the raw material to the product that has the highest sales margin, thereby maximizing the economic benefit. Even with a complex set of constraints, providing the problem could be formulated in monetary terms, the SOO approach as described by Equations (1)–(4) would in many cases be the normal method of solving this problem.

$$\text{Minimize or Maximize} F_1(x) \tag{1}$$

$$\text{Subject to}: \ x^{\text{L}} \leq x \leq x^{\text{U}} \tag{2}$$

$$g(x) \leq 0 \tag{3}$$

$$h(x) = 0 \tag{4}$$

Here, $F_1(x)$ is the objective or performance criterion or target to be minimized or maximized, $x$ is a vector of decision variables (also known as parameters), which can take any value between practical values selected as lower and upper limits of $x^{\text{L}}$ and $x^{\text{U}}$, respectively, and $g(x)$ and $h(x)$ are inequality and equality constraints, respectively. There can be none or several inequality and/or equality constraints. The optimal values of $x$ are denoted by $x^*$, and the optimal of the objective is $F_1(x^*)$.

Sometimes, there is more than one objective. For Vilfredo Pareto, it was the individuals that made up the population, as each individual would act independently to maximize their wealth. In CPE, MOO problems are ones that demonstrate a trade-off between at least two different aspects or objectives or performance criteria. Typical trade-offs are given in Table 1. In this table, aspect 1 decreases when aspect 2 increases, and vice versa.

**Table 1.** Typical trade-offs in process optimization. CPE: chemical process engineering.

| Trade-Off in CPE Problems | Aspect 1 | Aspect 2 |
|---|---|---|
| Efficiency in use of capital | Operating costs | Capital costs |
| Raw material efficiency | Revenue from sales | Capital costs |
| Environmental benefits | Harmful emissions | Capital costs |
| Process safety | Risk profile | Capital costs |
| Reliability | Plant downtime | Capital costs |

The MOO formulation is given by more than one objective function such as $F_1(x)$, $F_2(x)$, and $F_3(x)$. In these, $F_1$ may be minimized or maximized, $F_2$ minimized or maximized, and so on. Whereas a SOO solution yields a single value for each component of $x^*$, MOO often yields a series of values for each component of $x^*$, which are referred to as a set of non-dominated solutions or Pareto optimal solutions. Corresponding to each of these solutions, there will be one set of values for objectives such as $F_1(x^*)$, $F_2(x^*)$, and $F_3(x^*)$. A plot of these optimal values of objectives against each other in a two- or three-dimensional plot is referred to as a Pareto optimal front. For visualizing this front, more than one plot may be required if there are four or more objectives. Examples of MOO solutions are provided in Section 2 of this paper.

Where the two objectives were operating and capital costs, as proposed in the first example in Table 1, the MOO problem could be easily rewritten as an SOO problem by using a discount factor to discount future operating costs or revenue or by annualizing the capital cost, which is then added as a component of the operating costs. From the 1980s, commercial process modeling software such as ProII, Hysys, and Aspen Plus allow the calculation of operating costs from the use of raw materials and utilities. Capital cost estimation software such as Icarus or enterprise-based estimates could be generated. Convexity requires that there is only a single optimum within the solution space described by $g(x)$, $h(x)$ and the bounds for $x$, between $x^{\text{L}}$ and $x^{\text{U}}$, and also that the constraints $g(x)$ and $h(x)$ are

convex), Providing the problem could be written so that it satisfies the requirements for convexity, the SOO solution could be obtained directly either using an in-built SOO solver (such as in Aspen Plus) or by running the modeling software from within a mathematical optimization software platform such as Matlab.

The SOO approach to the optimization of both capital and operating costs relies on a method of weighting the two objectives using a financial discount factor. This discount factor is itself a variable, which leads to uncertainty in the solution. As an side, process economics textbooks in the 1980s recommended discount factors of around 20%, but 10% may be more suitable for the current time. This means that an SOO solution using a 20% discount factor would need to be re-optimized, but the MOO solutions would still be valid. Secondly, the need to satisfy convexity requirements also leads to an over-simplification of the original SOO problem. This is to avoid the chance of multiple optima and convergence to a local rather global solution.

The last two decades has seen a strong growth in the use of MOO as a valuable tool in CPE for process optimization. This is partly driven by the limitations of the SOO approach (such as where several objectives should be used for the application under study and results are for just one optimal solution), but also by the increasing aspects of process performance, which cannot be simply related to a monetary criterion. This is particularly related to sustainability (examples 3 and 4 in Table 1). Sustainability and sustainable development are central to the activities of most large corporations and local, regional, and national governments. Annual reporting, which includes environmental and social performance, is commonly referred to as a corporation's "Triple Bottom Line", and these are often published in a Sustainability Report. The fact that environmental and social performance cannot simply be weighted and added to produce a single target means that sustainability is essentially an MOO problem.

One of the most important concerns to chemical engineers at the start of the decade of the 2020s is global warming-driven climate change. Activities involving CPE result in close to 50% of all human-produced $CO_2$, the main exceptions being agriculture and land clearing. CPE process synthesis and design must weigh up economic performance against global warming potential, taking into account the whole life cycle rather than just a single process in isolation. Therefore, there is a need to widen the boundaries of process optimization. Whereas previously, it might have been around a single unit operation or a single plant, with global warming potential and other pervasive environmental problems, it is now necessary to calculate the results over the whole life cycle inventory, more than often using specialist Life Cycle Assessment software.

In the next section of this paper, the typical methodology of solving an MOO problem is described in detail. This is followed by an overview of MOO papers published in the field of CPE over the past 20 years. A short summary of some of these papers is given to demonstrate the breadth of the topics studied and approaches used. Both these are in Section 3. The objectives used in MOO applications in CPE are categorized and described in Section 4. Computational aspects of MOO, namely, techniques for solving MOO problems, Pareto ranking methods for selecting one of the Pareto optimal solutions, and available software for MOO are covered in Section 5. Finally, this paper ends with a detailed discussion on MOO, including similarities and differences between MOO and SOO, as well as suggestions for further research on MOO.

## 2. Procedure for MOO

Process optimization for one or more objectives requires a systematic procedure for meaningful, reliable, and useful results. A systematic procedure for MOO of processes consists of 5 steps, which are shown in Figure 1. As will be seen from the following description of the steps, 4 of the 5 steps require domain expertise (i.e., background and knowledge) in the application area besides some computational background. This requirement is often not recognized.

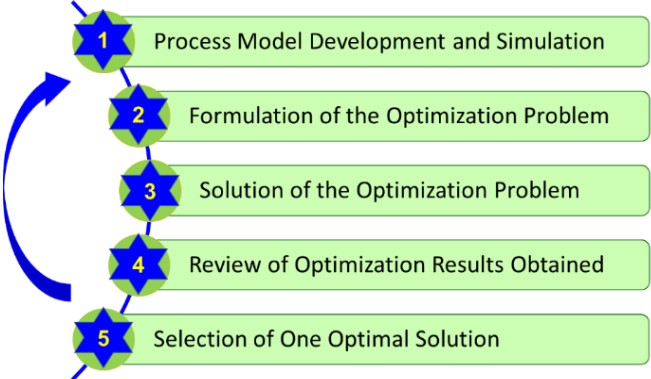

**Figure 1.** Steps in the systematic procedure for the multi-objective optimization (MOO) of processes; curved arrow on the left emphasizes the need for repetition from an earlier step.

**Step 1. Process Model Development and Simulation:** This is an essential and important step for any process optimization. The process model refers to a set of equations that correctly predict the response of the physical process under study. Equations can be algebraic and/or differential equations, and variables in them may be part of the system design and/or come from the operating conditions of the process. Examples of design variables are the reactor volume, heat exchanger area, number of stages in a distillation column, and membrane area in a membrane separation unit, whereas examples of operating variables are the reactor temperature, compressor outlet pressure, reflux ratio in a distillation column, and feed and permeate pressures in a membrane separation unit. Sometimes, design and operating variables together are referred to as design variables or parameters. Some of these variables (e.g., number of stages in a column) can be the discrete or integer type.

The model equations can be based on fundamental principles (such as mass and energy balances), in which case the model is known as a First Principles model. Such models may involve physical, chemical, thermodynamic, and/or transport properties of materials and streams in the process. Instead of developing the equations based on fundamental principles, a suitable simulator (e.g., Aspen Plus for simulating chemical processes) can be used. Such a simulation model is essentially a First Principles model except that governing equations are embedded within the simulator, and they are neither formulated by nor accessible to the user.

Alternately, the model equations can be based on the input–output data of the process obtained by experimentation on a physical setup or simulated experiments on a complex model (whose solution takes a long time and so the direct use of a complex model in optimization is not possible). Such models are referred to as empirical or response surface or surrogate models. Artificial neural networks (ANNs) is one commonly used method for generating the empirical models. Design of experiments can be used for obtaining efficiently the input–output data suitable for model development, which involves the regression of input–output data. In short, the two main types of models are First Principles and empirical models. A semi-empirical model is based on a combination of First Principles and input–output data.

After developing a suitable model (be it First Principles, in a simulator or empirical) of the process, it is important to confirm its accurate representation of the physical process. This testing requires finding and analyzing predictions (or responses or results) of the model for a range of situations (i.e., design and operating conditions). It may involve the numerical solution of governing equations (i.e., model simulation) successfully. If the model is developed in a simulator such as Aspen Plus, then the simulation should be performed without any error messages in the simulator for a range of situations. Thorough testing of the model not only confirms its accurate representation but also gives confidence that the process can be simulated successfully for numerous trial solutions generated during the search for the optimum.

Process modeling and simulation in the context of optimization are briefly covered by Sharma and Rangaiah [2]. See the book by Hangos and Cameron [3] for more details on process modeling and analysis of the resulting model.

**Step 2. Formulation of the Optimization Problem:** This step involves the following tasks: (a) selection of objectives and the development of equations for them; (b) decision variables and their bounds; and (c) identification of constraints and the development of equations for them. Objectives are criteria that quantify the performance of the process. They can be related to economics, energy consumption, environmental impact, controllability, safety, etc. MOO allows the inclusion of any number of objectives. The selection of objectives is according to the desired aspect of process performance. Obviously, more than one objective can be chosen in MOO, whereas SOO allows the selection of only one objective.

The selection of decision variables (sometimes known as parameters) requires in-depth knowledge of the process under study, and there can be many decision variables. The difficulty of solving an optimization problem increases with the number of decision variables. Hence, it is desirable to reduce the number of decision variables as far as possible while ensuring that the variables affecting the chosen objectives are not excluded. Sensitivity analysis (i.e., variation) of objectives to changes in design and operating conditions of the process is useful for choosing appropriate decision variables. This sensitivity analysis can be performed by changing one design/operating condition at a time, within a reasonable range of that variable. Where the effect of a variable is not well understood, factorial experiments can be performed on the model for changes in potential design/operating conditions. An advantage of factorial experiments over sensitivity analysis is that they can identify variables that are confounded.

Sensitivity analysis or factorial experiments provide not only a sensitivity of objectives with respect to design/operating variables, but also confirm successful solution of the model (i.e., simulation) of the process. Unsuccessful solutions at this stage in the procedure are useful so that the simulation can be made as robust as possible prior to launching the optimization. Having determined the effect of significant design/operating variables on the objectives, these variables can be chosen as decision variables. Lower and upper bounds on each decision variable can be decided based on process knowledge in conjunction with the results from the sensitivity analysis. The results of the sensitivity analysis can also be used to choose only those objectives that show variation with changes in decision variables.

Constraints in an optimization problem can be equality and/or inequality type. These depend on the process to keep it within a feasible and safe operating range. When the process model is built using a simulator such as Aspen Plus, only several inequality constraints are likely to be involved in the optimization problem. This is because many governing equations arising from mass, energy, and other balances are embedded within the simulator, and they are satisfied when the process is successfully simulated by the software. If a First Principles model is used, then the optimization problem includes the governing equations as equality constraints. Relevant decision variables, their bounds, and their constraints should be chosen carefully in order to achieve meaningful results from MOO.

**Step 3. Solution of the Optimization Problem:** This step requires the selection and use of a MOO technique for solving the formulated problem. As outlined in a later section, many programs for MOO are now available, and some of them are free. So, there is no need to develop a program for solving the formulated problem unless a new and/or improved technique for MOO is being studied. However, there may be a need to interface the available MOO program in one platform (e.g., MS Excel or Matlab) with the simulation model on another platform (e.g., Aspen Plus). The interfacing of Aspen Plus with Matlab and Aspen Hysys with MS Excel is described with an example in [4] and [2], respectively.

The following is a list of points of advice for correctly using any of the MOO software.

- Carefully read and follow the instructions that come with the MOO program.
- Learn how to use and also test the MOO program before using it for any application for the first time. For testing, choose a mathematical optimization problem with a known solution from the

　　　　literature and/or the example provided with the MOO program and reproduce the known optimal solutions using the MOO program with default values in it for algorithm parameters.

- Be careful in correctly providing/entering the required inputs such as objectives, decision variables, bounds, and constraints of the application problem to the MOO program. Any wrong inputs may lead to failure of the program or incorrect results from the program. First, test the MOO software with a relatively narrow range of decision variables around a known solution. If successful, the range of decision variables can be widened as required.

　　　　**Step 4. Review of Optimization Results Obtained:** Two or more objectives in the formulated MOO problem are likely to be conflicting (i.e., improvement in one objective is accompanied by the worsening of another objective). Hence, unlike one or a few optimal solutions from SOO, MOO gives many optimal solutions that are known as Pareto optimal front (or solutions) and as non-dominated solutions. Results from MOO can be presented in three spaces, namely (1) objective space, (2) decision variable space, and (3) objective versus decision variable space; the first two spaces can be in more than two dimensions, depending on the number of objectives and decision variables. Objective space plots are not relevant in the case of SOO, since it involves only one objective.

　　　　Figure 2 illustrates the results of optimizing a membrane process for $CO_2$ removal from natural gas. Figure 2a depicts the optimal trade-off of the two objectives: maximize methane purity and maximize methane recovery, simultaneously. Figure 2b is the plot of decision variables—membrane 1 area and membrane 2 area—in the decision variable space. It shows that the optimal value of the membrane 1 area is at its upper bound of 10,000 $m^2$, which means that both objectives improve if this bound is increased further, but within a practical limit. On the other hand, the optimal value of the membrane 2 area varies from a low value (but above its lower bound) to its upper bound of 10,000 $m^2$. The variation of objectives with decision variables is presented in Figure 2c–f. For example, Figure 2c is a plot of methane purity versus the membrane 1 area. Figure 2c,d indicates that the membrane 1 area affects both objectives in the same direction, whereas Figure 2e,f show that the increase in the membrane 2 area increases methane purity but decreases methane recovery.

　　　　After obtaining results from solving the MOO problem, plots such as those in Figure 2 should be created; use the lower and upper bounds of a decision variable for the axis scale (and not an expanded scale) so that the extent of its variation can be seen easily. Review the Pareto optimal front in Figure 2a for its spread, trend, smoothness, and discontinuities. Depending on the MOO problem size and MOO algorithm parameters (e.g., population size and maximum number of generations), the Pareto optimal front may not be as smooth as in Figure 2a. Next, find a qualitative explanation based on underlying principles of the application, for each variation between objective and a decision variable such as that in Figure 2c–f.

　　　　The MOO problem and its results are satisfactory if the Pareto optimal front is sufficiently wide and smooth, and the variation of objectives with decision variables can be explained based on the underlying principles of the application. If not, check the following: the adequacy of the model developed in Step 1 and used in the MOO problem; the optimization problem formulated (i.e., objectives, decision variables, and their bounds, and constraints) in Step 2; inputs provided to the MOO problem; and the MOO algorithm parameters used. Common MOO algorithms are briefly described later in Section 5.

　　　　If the MOO algorithm is a stochastic technique (i.e., uses random numbers during its search), which is also known as metaheuristics and includes evolutionary algorithms, then the optimal solutions found depend on random numbers, and they may be approximate. Hence, it is advisable to solve the MOO problem using the same program but with a different initiator for random numbers for better results and/or confirmation. Optimal solutions found from such multiple runs can be combined and sorted for non-domination; see the book chapter by Sharma et al. [5] for more details including an MS Excel program for this sorting.

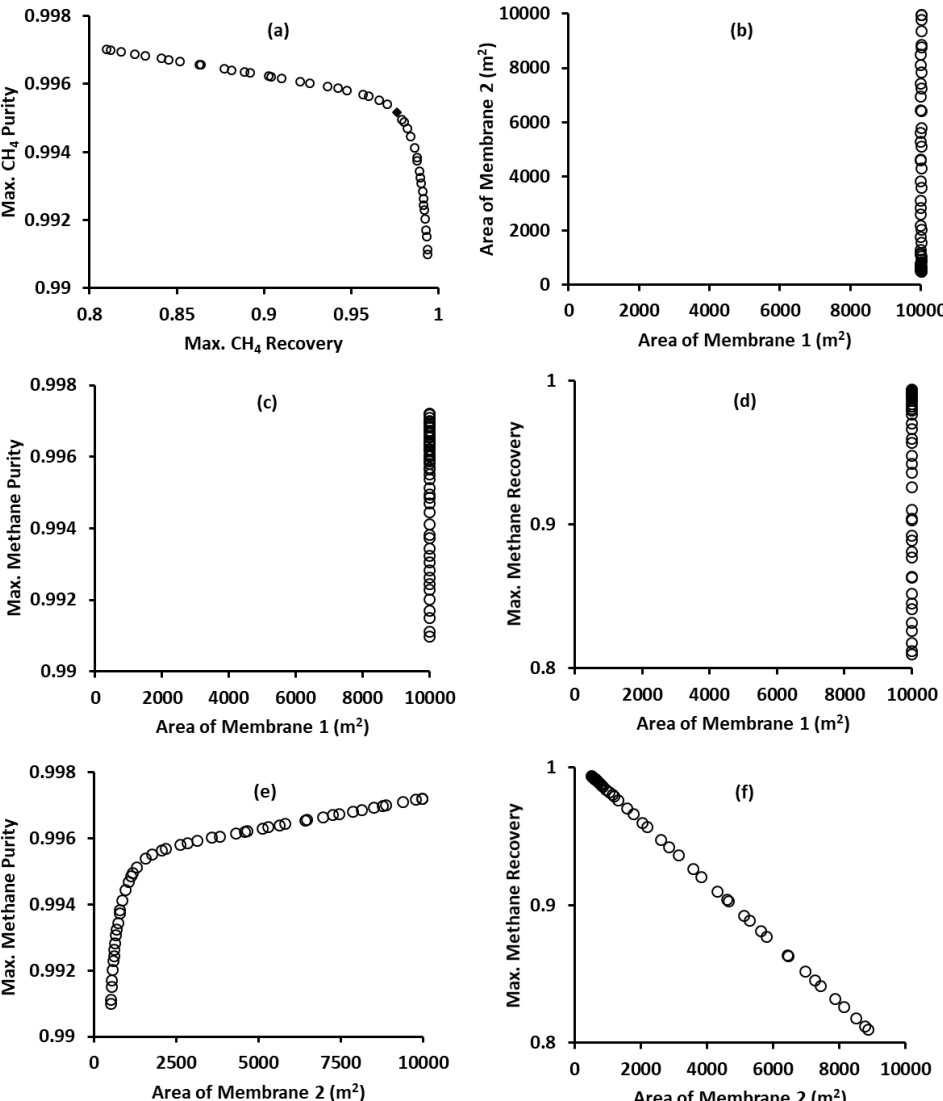

**Figure 2.** Pareto optimal (non-dominated) solutions for optimizing a two-stage membrane separation process for $CO_2$ removal from natural gas: (**a**) objective space showing the trade-off between two objectives, (**b**) decision variable space showing optimal values of decision variables, and (**c**) to (**f**) variation of one objective function (methane purity or recovery) with decision variables (membrane 1 and 2 areas).

Thus, after reviewing the optimal solutions obtained (i.e., Step 4), some steps in the MOO procedure in Figure 1 may have to be repeated from Step 1, 2, or 3. The curved arrow in Figure 1 signifies this type of iteration.

**Step 5. Selection of One Optimal Solution:** As noted above and assuming some conflict among the objectives used, MOO gives many non-dominated (Pareto optimal) solutions, which are equally good from the point of the objectives employed. However, only one of these solutions is required for implementation for the application under study. Hence, this final step in MOO involves the selection of one of the non-dominated solutions for implementation. Note that this selection is performed in Step 5 after reviewing the obtained non-dominated solutions in Step 4. It should not be confused with the selection operation used in evolutionary algorithms such as genetic algorithms and differential evolution. Some MOO methods (e.g., the weighted sum technique of SOO approach described later) find only one Pareto optimal solution. In such a case, this step of selection is not needed; however, such MOO methods need inputs on the relative importance of objectives in the earlier Step 3 itself.

As can be seen from the reviews by Bhaskar et al. [6], Masuduzzamn and Rangaiah [7], and Sharma and Rangaiah [8], studies on MOO applications in CPE are focused on formulating and solving the MOO problem (i.e., Steps 1 to 4 in Figure 1) and not on Step 5. Possible reasons for this are as follows: (a) novelty and substantial work involved in formulating and solving the MOO problem; (b) perception that selection can be made based on engineering experience and preference; and (c) selection is not needed in the case of SOO. For example, among the non-dominated solutions in Figure 2a, an optimal solution (with methane purity of ≈0.995 and recovery of ≈0.97) in the region, where the slope of the Pareto optimal front changes significantly, is better for implementation, since this solution has values of both objectives close to the corresponding best value with only a small compromise in both objectives. The review by Rangaiah et al. [9] found that 20 out of 65 studies have used one or more selection methods, which shows an increasing awareness and application of selection methods in ChE.

In fact, many methods/techniques for selecting the preferred solution or for ranking the non-dominated solutions have been proposed and discussed in the area known as multi-criteria analysis (MCA), multi-criteria decision analysis (MCDA), or multi-criteria/choice decision making (MCDM). From now on, this area and the methods are referred to as MCA and MCA methods, respectively. The focus of this area is on the analysis or ranking of Pareto optimal solutions (also referred to as decision options), i.e., only Step 5 after finding/having Pareto optimal solutions, and not on all steps in MOO in Figure 1. Note that decision options are entirely different from decision variables in an optimization problem. The scope of MOO is much wider than that of MCA, and MOO can be referred to as multi-criteria optimization (MCO) but not as MCA/MCDA/MCDM.

MCA methods may require only the values of objectives from the Pareto optimal front (i.e., objective space) and not optimal values of decision variables (i.e., decision variable space). The matrix of values of n objectives at m Pareto optimal solutions is known as the decision or evaluation matrix. Here, this matrix will be referred to as the objective matrix, since objectives are commonly used in CPE applications of optimization. After having this matrix, MCA has the following 5 sub-steps:

(i) normalization of values of each objective;

(ii) choosing the weight for each objective;

(iii) use of the MCA method for ranking or scoring of m Pareto optimal solutions;

(iv) sensitivity analysis of Pareto ranking (i.e., ranking of Pareto optimal solutions) to normalization, weights, and/or MCA method used, and

(v) choose the top-ranking solution for implementation.

The above sub-steps are outlined below. The first sub-step of normalization of values of each objective is important, since the magnitude and range of each objective can be very different. There are at least 4 methods for normalization [10], wherein values of each objective are normalized using (1) the maximum value of that objective, (2) the sum of all values of that objective, (3) the difference between the maximum and minimum values of that objective, and (4) the square root of the sum of squares of all values of that objective.

In the second sub-step, the weight (i.e., relative importance) for each objective is chosen; the sum of weights for all n objectives must be unity. Weights can be given by one or more decision makers (i.e., managers and/or senior engineers). Alternately, they can be computed using one of many available methods, each using certain reasoning. These methods can be categorized into two groups: objective methods (which use an objective matrix and do not need any user inputs) and subjective methods (which require user inputs and do not need an objective matrix). A simple weighting method (mean weight) assigns the same weightage for all objectives (i.e., equal importance). More details on weighting methods and their effect can be found in [11,12].

The third sub-step is the application of an MCA method along with the weights given/found in the second sub-step to the normalized objective matrix, to rank Pareto optimal solutions and choose the top-ranked solution. Numerous methods for MCA have been proposed and studied in the literature. One MCA method, namely, simple additive weighting (SAW), is based on the sum of products of

normalized objectives with corresponding weights. One of the earliest and popular methods is the Technique for Order of Preference by Similarity to Ideal Solution (TOPSIS). It uses a positive ideal solution (PIS) having the best values for all objectives, and negative ideal solution (NIS) having the worst values for all objectives. Note that PIS is not achievable. The best or recommended solution according to TOPSIS is that having the smallest Euclidean distance from PIS and also the longest Euclidean distance from NIS. Note that these distances are in the objective space. Another MCA method, namely, Gray Relational Analysis (GRA), is interesting in that it does not require weights or any inputs from the user. It employs a gray relational coefficient to quantify the similarity between a Pareto optimal solution and PIS. More details on MCA methods and their application can be found elsewhere [10,12,13].

The fourth sub-step of sensitivity analysis of Pareto ranking to normalization, weights, and/or the MCA method used is necessary because of many possibilities for normalization, weights, and the MCA method. In addition to the results of sensitivity analysis, we assess the practicality of optimal values of decision variables (e.g., for safe and smooth plant operation) for implementation, for finalizing one of the optimal solutions. The optimal values can be corresponding to a few top-ranked Pareto optimal solutions, and the assessment of practicality can be qualitative or quantitative.

*Summary*: The systematic application of MOO in CPE (in fact, any field) involves 5 steps in Figure 1. The first 4 steps should be carried out sequentially before repeating one or more of them, as required. Successful completion of the first 4 steps gives the Pareto optimal front (non-dominated solutions), which provides insights into quantitative trade-offs among different objectives used. In Step 5, one of the Pareto optimal solutions is chosen with or without inputs from decision maker(s) for implementation. With the exception of Step 3, the other four steps require knowledge and expertise in the domain of the application. Further, some computational background is necessary in all but Step 4 for a review of the optimal solutions found. The readily available software for MOO and Pareto ranking is covered later in Section 5.3.

## 3. MOO Application in CPE

MOO applications in CPE have been reviewed in a few previous works [6–9]. A review of MOO techniques and applications in energy saving and emissions reductions is available in [14]. Recently, Madoumier et al. [15] reviewed MOO in food processes; their critical and detailed review includes objectives (performance indicators), a process model, and an assessment of trade-off in the context of food processes. Readers interested in a specific application of MOO in CPE can refer to one of these reviews and/or search in databases such as Scopus and Google Scholar for recent studies. They can also undertake a citation search of relevant papers found in the above reviews. Hence, the twin aims of this section are (a) to discuss the increasing role of MOO in CPE (by presenting number of journal papers, top contributors, journals, etc.) and (b) to outline selected recent papers as representative of diverse MOO applications in CPE.

### 3.1. Detecting an Increasing Role of MOO in ChE

To detect the increasing role of MOO, the Scopus database is searched for journal articles or reviews (collectively referred to as papers from now on) containing phrases: (multi and objective and (optimization OR optimisation)) OR (bi and objective and (optimization OR optimisation)), in the title, abstract, and keywords. Further, this search was defined as from year 2000 (i.e., after 1999) to the end of October 2019, and it was also limited to the subject area of ChE and the English language. It was observed that all journals, as included in the subject area of ChE in the Scopus database, may not be of interest to ChE researchers. Hence, the search was limited to only those journals deemed more related to CPE and with at least 4 relevant papers during the period of search. In all, 1500 papers were found in 70 different journals. Among these journals, those with at least 40 relevant papers are given in Table 2. There may be papers using MOO in their study but without the words used in the above search. For example, Da Cunha et al. [16,17] employed MOO in their study on retrofitting formic acid

process, but these papers are not captured in the above search. For this and because of the journal selection stated above, the number of journal papers on MOO applications in CPE from the year 2000 to October 2019 is expected to be more than those presented in Table 2 and Figure 3.

**Table 2.** Titles of journals (which published at least 40 relevant papers) and number of papers in each of them from the year 2000 to October 2019: subject area of chemical engineering (ChE) in Scopus.

| Title of Journals | Number of Papers |
| --- | --- |
| Computers and Chemical Engineering | 247 |
| Computer Aided Chemical Engineering | 90 |
| Industrial and Engineering Chemistry Research | 85 |
| Applied Sciences (Switzerland) | 80 |
| International Journal of Heat and Mass Transfer | 68 |
| Chemical Engineering Science | 64 |
| Chemical Engineering Research and Design | 63 |
| Chemical Engineering Transactions | 52 |
| Desalination | 47 |
| AIChE Journal | 41 |

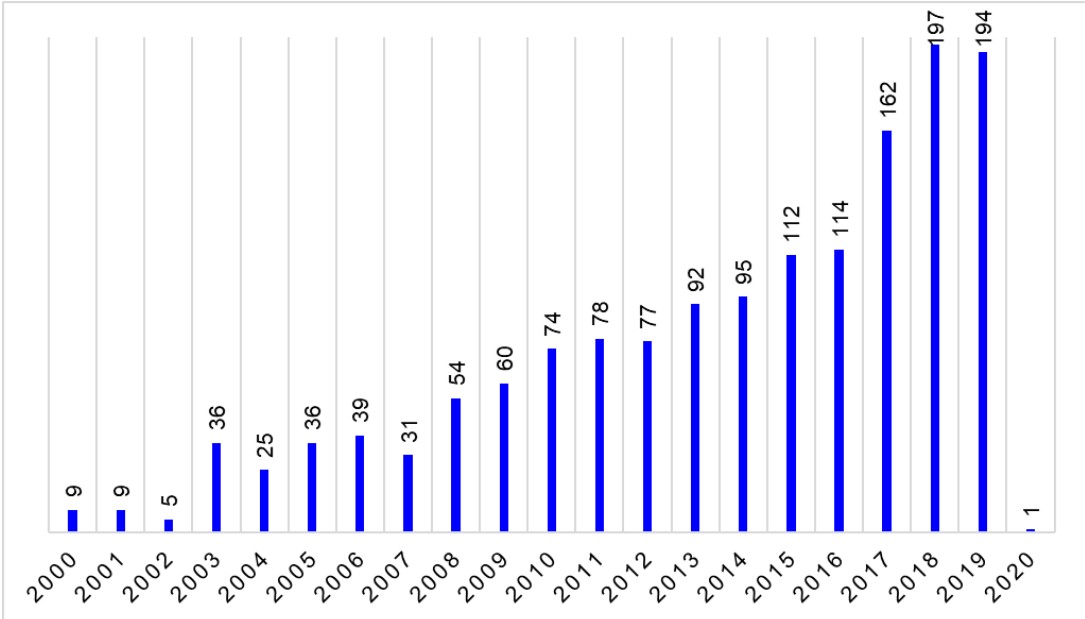

**Figure 3.** Number of journal papers containing multi/bi and objective and optimization/optimisation in the title, abstract, and keywords, from the year 2000 to the end of October 2019 in the journals related to CPE in the subject area of Chemical Engineering in Scopus. The number for the year 2019 is for 10 months only, and there is one paper for the year 2020.

Table 2 shows that the highest number of papers (247) on MOO applications in CPE were published in the Computers and Chemical Engineering. This is not surprising considering the computational orientation of this journal, as clearly indicated by its title. It also emphasizes the computational nature of MOO applications in CPE. The next six places for reported MOO applications (with 63 to 90 papers) are taken up by journals with a greater focus on process applications; of these, Computer Aided Chemical Engineering is a book series based on presentations in international conferences related to computational studies in CPE. The total number of journals shown in Table 2 is 27. It is of interest to note that 17 papers on MOO applications were published in the open access journal: Processes, which started in the year 2013 only (i.e., about 3 papers annually).

The increasing role of MOO in CPE can be seen by the growth of journal papers found by the search stated above over the last 20 years, as shown in Figure 3. The total number of papers covered

in this figure is 1500, of which 159 are in open access journals and 10 are articles in press. Starting from less than 10 papers in 2000 to 2002, the number of papers increased to about 35 in 2003 to 2007, to around 80 in 2010 to 2014, and nearly 200 in 2018; the number of papers in 2019 is expected to be more than 200 (Figure 3).

Figure 4 shows that there are 24 researchers, each of whom contributed 10 or more journal papers to MOO and its applications in CPE, from the year 2000 to the end of October 2019. Of them, the top 2 researchers (co-)authored 29 and 28 papers, and each of the next 4 researchers contributed 21 to 24 papers. Among the researchers in Figure 4, G.P. Rangaiah and K. Hidajat have been with the National University of Singapore (NUS), A.K. Ray was with NUS before he moved to the University of Western Ontario, and S.K. Gupta was also with NUS for two years on leave from Indian Institute of Technology (IIT) Kanpur.

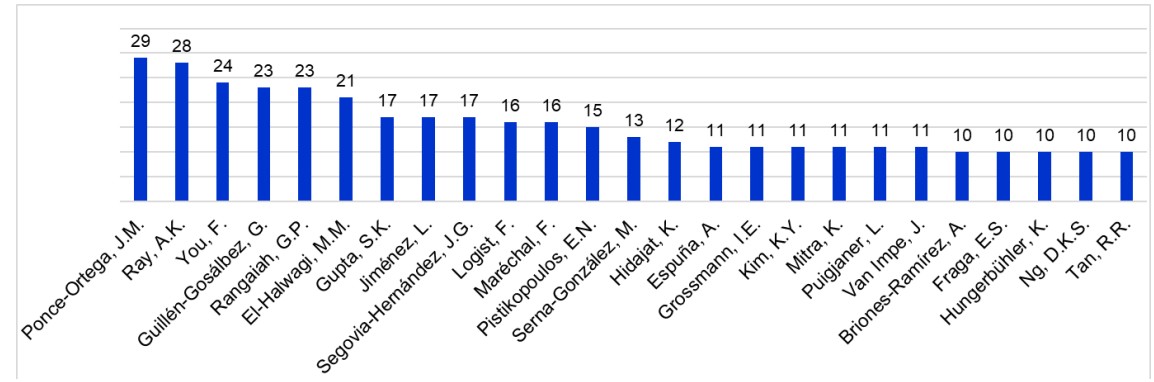

**Figure 4.** Researchers who (co-)authored 10 or more papers in Figure 3.

Further scrutiny of Table 2 indicates that the journals classified in the subject area of ChE in Scopus do not cover all journals wherein ChE academicians and practitioners publish their research and development findings; such journals can be seen in Table 3. Hence, the Scopus database was searched for journal articles or reviews containing phrases: ((multi and objective and (optimization OR optimisation)) OR (bi and objective and (optimization OR optimisation)) in the title, abstract, and keywords. This new search was limited to journals perceived to be of interest to Chemical Engineers within the subject area of Energy (instead of ChE chosen for the previous search) and with at least 4 relevant papers during the period of search but excluding the journals in Table 2 to avoid duplication with the previous search. As before, this search was limited to papers from the year 2000 (i.e., after 1999) to end of October 2019, and the English language.

**Table 3.** Titles of journals (which published at least 15 relevant papers) and number of papers in each of them in the subject area of energy in Scopus from the year 2000 to October 2019.

| Title of Journals | Number of Papers |
|---|---|
| Applied Energy | 366 |
| Energy Conversion and Management | 339 |
| Journal of Cleaner Production | 310 |
| Energies | 279 |
| Applied Thermal Engineering | 236 |
| Renewable Energy | 143 |
| Sustainability (Switzerland) | 123 |
| Energy | 96 |
| International Journal of Hydrogen Energy | 76 |
| Renewable and Sustainable Energy Reviews | 65 |
| Solar Energy | 58 |
| Journal of Renewable and Sustainable Energy | 49 |

The total number of papers found by the new search is 2744 in 58 different journals. This is almost double the papers in CPE journals. Among these journals, those with at least 40 relevant papers are presented in Table 3. These journals and papers found by the search may be of interest to some academicians and practitioners in ChE. Of the 2744 papers, 500 are in open access, 21 are articles in press, 1892 are published in the first 8 journals in Table 3, and the first 3 journals (with 366 to 310 papers) account for 1015 papers. The fourth journal in this table is Energies, which is an open access journal that has been published since 2008. Even then, it has published 279 papers on MOO applications relating to energy in just 12 years.

Figure 5 indicates an exponential rise in the number of journal papers containing multi/bi and objective and optimization/optimisation in the title/abstract/keywords from just a few in the year 2000, 62 in the year 2010, to 518 in the year 2018. This number for the year 2019 is likely to exceed 700. Figure 6 shows that there are 18 researchers, each of whom contributed 10 or more papers found in the search. Of them, the top four researchers (co-)authored 27 to 24 papers, and each of the next two researchers 18 papers. A few of these researchers contributed to many papers in the ChE subject area as well (Figure 4). These are not repetition, since the journals chosen in the ChE subject area (Table 2) are different from those in the Energy subject area (Table 3).

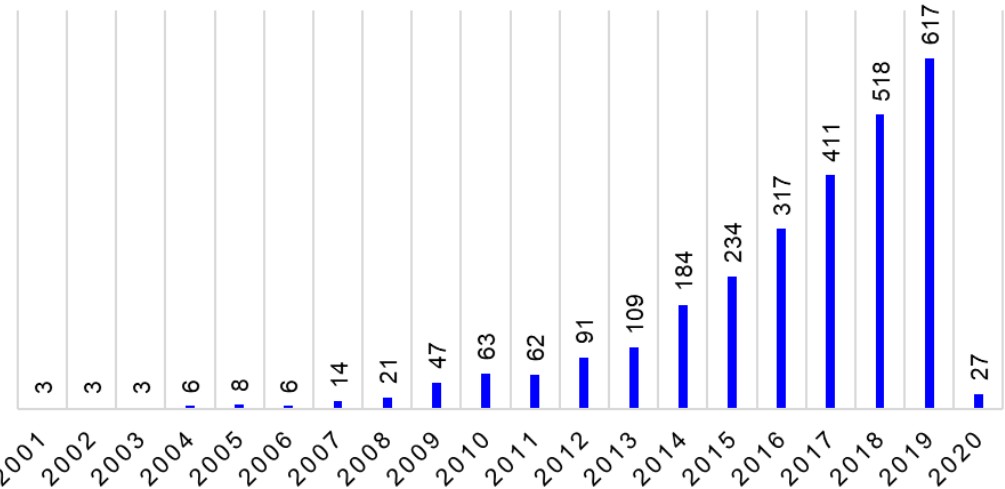

**Figure 5.** Number of journal papers containing multi/bi and objective and optimization/optimisation in the title, abstract, and keywords, from the year 2000 to the end of October 2019, in the journals likely of interest to chemical engineers in the subject area of energy in Scopus. The number for the year 2019 is for 10 months only, and there are already 27 papers for the year 2020.

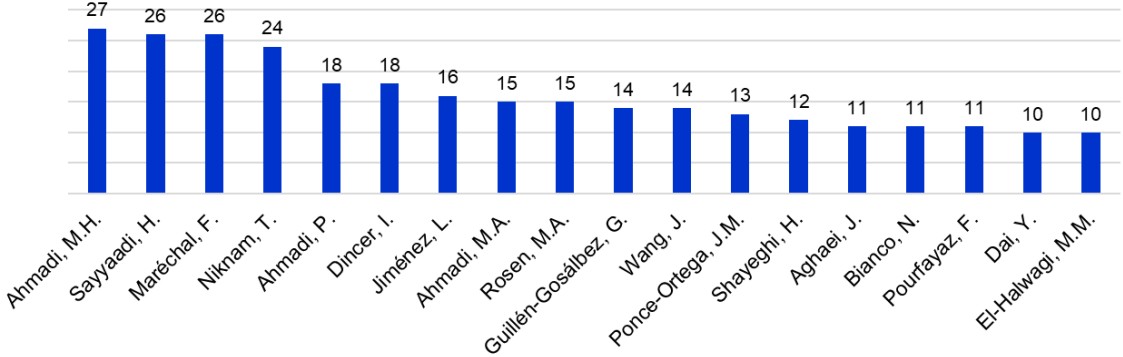

**Figure 6.** Researchers who (co-)authored 10 or more journal papers/reviews in Figure 5.

***In summary***, the number of papers with multi/bi and objective and optimization/optimisation in the title, abstract, and keywords in the journals of interest to ChE academicians and practitioners (within the subject areas of ChE and Energy in Scopus) from the year 2000 to the end of October 2019 is

more than 4200. They are contributed by numerous researchers from 160 institutions in 90 different countries. A steady and significant increase in the number of papers on MOO applications in CPE can be clearly seen in Figures 3 and 5. Hence, it can be concluded that MOO has been extensively employed for CPE applications and that the majority of future studies on optimization will be for two or more objectives.

Although the two searches presented above covered many journals, there are some other journals of interest to CPE researchers. A few examples of such journals are *Minerals and Metallurgical Processing*, *The International Journal of Mineral Processing*, and *Macromolecular Theory and Simulations*. These have not been included in these statistics.

## 3.2. Selected Applications of MOO in ChE

MOO in ChE covers a wide range of applications such as for designing a new process, for retrofitting/revamping an existing process, for process control, for planning and scheduling, for supply chain optimization, for parameter estimation, and for product design. There have been many MOO studies that fall into each of these categories, and it is not possible to review all of them in this paper. Hence, only several recent papers covering these categories from top researchers in Figure 4 are outlined in this section.

Cabrera-Ruiz et al. [18] studied both the design and controllability of intensified distillation sequences by MOO for minimizing the total annual cost (that includes both operating cost and annualized capital cost) and controllability index (based on the area under the curve of condition number versus frequency). The cases studied are (a) two distillation columns with heat integration and vapor recompression, (b) a conventional reactive distillation sequence, and (c) a reactive dividing-wall column. Differential evolution with tabu list (DETL) was used for solving the MOO problems in these case studies.

Fuentes-Cortes et al. [19] used MOO for the optimal design of a combined heat and power system for meeting the needs of a housing complex of more than 1400 households. The system based on biogas and natural gas includes an internal combustion engine, microturbine, fuel cell, Stirling engine, and thermal storage. The MOO problem was solved using the weighted sum technique and GAMS programs. It provides a wide range of trade-off solutions between cost and greenhouse gas emissions that can stimulate development of local biogas markets.

Manesso et al. [20] proposed MOO for the model-based design of experiments for estimating more accurate parameters. They suggested a Fisher information matrix and model curvature as the two objectives and illustrated the proposed approach for estimating kinetic parameters from experiments on a fed-batch fermentation of Baker's yeast. The bi-objective problem was solved using NSGA-II in MATLAB,

Ooi et al. [21] proposed a multi-objective framework for molecular design and illustrated it for the solvent design for oil extraction from palm-pressed fiber. The target properties (i.e., objectives) considered are several physical properties (related to product functionality) along with a number of environmental/safety aspects. The framework consists of an analytic hierarchy process involving the pairwise comparison of chosen objectives to determine the weights for each of the objectives; then, the resulting SOO problem was solved to find one Pareto optimal solution.

Palys et al. [22] utilized MOO for the performance improvement of a packed bed membrane reactor for the production of ethylene oxide from ethylene. They studied several cases of bi-objective optimization of both the operation and design of this reactor. The two objectives used in each case are chosen from conversion of ethane, the selectivity of ethylene oxide, and the flowrate of ethylene oxide. Real coded NSGA-II was employed for solving the bi-objective optimization problems.

Beykal et al. [23] optimized the energy system for a commercial building (supermarket), for minimizing cost and $CO_2$ emissions. The energy system consists of several energy sources including biomass and natural gas for the on-site generation of electricity and heat required in the building. The framework employed for this complex case study has three main stages of sampling,

surrogate modeling, and optimization. Beykal et al. [23] found many Pareto optimal solutions for the bi-objective optimization problem of minimizing cost and $CO_2$ emissions by solving a series of SOO problems formulated using the ε-constraint technique.

Gong and You [24] employed the ε-constraint technique for the complex, multi-level optimization problem of designing resilient process systems. The minimization of total capital cost and maximization of resilience (toward major disruption events) are the two objectives in the bi-objective problem. They applied the proposed framework to the design of two process networks: one consists of a number of petrochemical processes, and another is shale gas processing and the recovery of natural gas liquids (NGLs).

Kundu et al. [25] investigated the potential of MOO for parameter estimation in sorption and phase equilibria problems. One objective is for the error in fitting vapor–liquid equilibrium data and another for the error in fitting the heat of mixing (excess enthalpy) data. They found that the MOO problem solution by a genetic algorithm provides a more uniform and wider Pareto optimal front compared to solving the equivalent SOO problem via the weighted sum technique.

Sharma and Marechal [26] studied the design of solid oxide fuel cells with gas turbine hybrid system for producing heat and electricity as well as maximizing profit while minimizing annualized capital cost. They performed both normal MOO and robust MOO (to include parametric uncertainties) by a genetic algorithm combined with a clustering technique and then compared the resulting Pareto optimal solutions for normal and robust MOO.

Wheeler et al. [27] optimized the design of biomass to a biofuels supply chain for 11 environmental criteria along with one economic criterion (net present value). Their approach is to determine weights for each criterion based on surveys of stakeholders, convert the MOO problem into an SOO problem using the weighted sum technique, and then solve the resulting SOO problem. Such an approach is computationally efficient, particularly when there are many objectives (12 in the application studied), but it requires a priori surveys to estimate weights and gives only one Pareto optimal solution corresponding to the weights used.

Mano et al. [28] optimized the synthesis of heat exchanger networks for cost (= sum of annualized capital cost and utility cost) and four environmental objectives (namely, impact on human health, eco-systems quality, resources depletion, and global warming). Survey responses from experts and the fuzzy analytic hierarchy process were employed to find the weights to convert the MOO problem with five objectives into an SOO problem, which was then solved using GAMS.

Nimmegeers et al. [29] studied the robust dynamic optimization of a fed-batch fermenter (bioreactor) for lysine production. They employed the sigma point approach (unscented transformation) to account for parametric uncertainty and ParetoBrowser for the interactive MOO of two conflicting objectives, namely, yield, and productivity.

Panu et al. [30] employed bi-objective optimization for the design of carbon–hydrogen–oxygen symbiosis networks to minimize both cost and $CO_2$ emissions. The purpose of this network is to improve the integration of existing plants by adding new plants for capturing and converting $CO_2$ into the chemicals used by the existing plants. In a way, this is revamping an existing eco-industrial park by the addition of new plants. Panu et al. [30] found a number of trade-off solutions between cost and $CO_2$ emissions using the ε-constraint technique.

Punase et al. [31] optimized a fixed bed catalytic reactor for hydrogen production from ethanol by steam reforming for simultaneously maximizing the hydrogen mole fraction and minimizing the sum of CO and $CO_2$ mole fractions, using NSGA-II. Before optimization, they estimated the kinetic parameters by fitting experimental data using a genetic algorithm and found them to be better than the two sets of values reported in the literature.

***In short***, there is a wide range of MOO applications in ChE, and the MOO techniques used for solving these application problems include evolutionary algorithms (metaheuristics), the ε-constraint technique, and the weight sum technique. These MOO techniques are elaborated later in Section 5.

## 4. Objectives Used for MOO

As noted in the previous section, MOO was applied to numerous applications in CPE. These applications are for designing a new process, for retrofitting/revamping an existing process, for process control, for planning and scheduling, for supply chain optimization, for parameter estimation, and/or for product design. Not surprisingly, they employed different objectives or performance criteria appropriate to the respective application. In this section, the commonly used objectives in CPE applications of MOO are reviewed in the following categories: Fundamental Criteria, Economic Criteria, Energy Criteria, Environmental Criteria, Control Criteria, and Other Criteria. Here, the word criteria is used, and it has the same meaning and significance as objectives in the optimization literature.

Criteria used in the MOO of CPE applications are often from two or more different categories described below (e.g., one from an economic type and another from an environmental category). In some applications, they are from the same category (e.g., conversion and selectivity from the fundamental category, and capital and operating costs from the economic category). In either case, the chosen criteria are usually conflicting either fully or partially. However, conflicting criteria are not necessary for MOO. If the chosen criteria are not conflicting, then MOO will give a single optimal solution, which is the best for all criteria used, and not a set of non-dominating (Pareto optimal) solutions. It is desirable to employ conflicting criteria in MOO in order to obtain a range of optimal solutions and thus deeper insights into the application under study.

**Fundamental Criteria:** The fundamental criteria often relate to the physical and/or chemical performance of a reaction or material to quantify the performance of a reaction or separation process; some examples of fundamental criteria are the conversion, selectivity, recovery rate, product purity, production rate, etc. The conversion of a reactant is the ratio of the feed converted to the total feed used, whereas the selectivity of a product is the ratio of the desired product formed to the total reactant converted multiplied by the stochiometric factor [32]. Both of these are important to quantify the performance of an industrial reactor wherein a number of reactions are occurring. They have been employed to optimize the design of reactors such as steam methane reformer [33], a stirred tank reactor for simultaneously producing ethyl tert-butyl ether and tert-amyl ethyl ether using response surface methodology [34], and a fixed-bed reactor of methanol oxidation to formaldehyde [35].

The recovery rate is another commonly used objective to maximize the ratio of the quantity of the product to that in the feed/inlet stream. It is often employed along with other objectives such as product purity, production rate, economic, and/or environmental objectives, to optimize the process design. Beck et al. [36] applied MOO to identify the trade-off between $CO_2$ product purity, recovery rate, and power consumption in the design of a vacuum/pressure swing adsorption process for separating $CO_2$ from flue gas. Estupiñan Perez et al. [37] used $CO_2$ product purity and the recovery rate as two objectives of MOO to validate experimentally the designed vacuum swing adsorption process for $CO_2$ capture. Reddy et al. [38] chose product yield and batch time as the objectives in the optimal design of a reactive batch distillation process. Yasari [39] applied the production rate and catalyst deactivation rate in the dynamic optimization of two types of tubular reactors for producing dimethyl ether. You et al. [40] defined the separation efficiency indicator to optimize extractive distillation processes; this particular indicator quantifies the ability of the extractive section to discriminate the desired product between the top and the bottom parts of the extractive section.

**Economic Criteria:** As expected, these criteria are often used objectives in the optimization of CPE applications, to design the process system having the most economic viability. The total capital cost (TCC) and annual operating cost (AOC) are the two important economic criteria for designing a new system or retrofitting/revamping an existing system. The former refers to the fixed cost for setting up a new system or modifying an existing system. TCC is often estimated using capital cost correlations, which are functions of the size and operating conditions of the equipment. The installed cost (and not purchase cost) of equipment must be used for a realistic estimation of TCC, since the installed cost can

be 1.5 to 4 times the purchase cost (e.g., see Table 16.11 in the book by Seider et al. [41]). AOC refers to the sum of variable costs such as steam, coolant, electricity, raw material, and catalyst costs.

The widely used correlations for estimating TCC in CPE were from the books by Seider et al. [41], Douglas [42] and Turton et al. [43]. A simple expression for combining TCC and AOC into total annual cost (TAC) is defined as the sum of annualized TCC (using the payback period, PBP) and AOC [44].

$$\text{TAC} = \frac{\text{TCC}}{\text{payback period}} + \text{AOC} \tag{5}$$

Here, PBP is the time required, after the start-up of the system, to recover the TCC spent for setting up the system. It is in the range of 3 to 5 years. Instead of PBP, some researchers used the plant lifetime (such as 10 or 15 years), which underestimates the contribution of TCC to TAC. Plant lifetime may be reasonable for depreciation calculations but not acceptable for annualizing TCC, which should include equipment maintenance costs and interest on borrowed capital besides depreciation. Hence, the TAC calculation should be based on 3 to 5 years for PBP.

In recent years, Da Cunha et al. [16,17] employed AOC and TCC as the two objectives for optimizing the design and then the retrofit/revamp of a formic acid production process. Lee et al. [45] used TAC and an Eco-indicator 99 to design a sustainable carbon capture and storage infrastructure under uncertainty. Ma et al. [46] conducted a tri-objective optimization for designing an exergy-saving thermally coupled ternary extractive distillation process using TAC, $CO_2$ emission, and thermodynamic efficiency as the objectives. Shang et al. [47] conducted a tri-objective optimization with TAC, $CO_2$ emission, and separation efficiency as the objectives to optimize an extractive distillation for separating ethanol–water mixture using a deep eutectic as the entrainer. TAC as an economic objective, Eco-indicator 99 (ECO99) as an environmental objective, and Individual Risk (IR) as a safety criterion have been used in the MOO of alternate reactive distillation processes for ethyl levulinate production [48].

Furthermore, profitability analysis can be implemented based on the estimated TCC and AOC using the procedures in the books by Seider et al. [41] and Turton et al. [43]. For example, Patle et al. [49] optimized two alternate alkali catalyzed processes for producing biodiesel from waste cooking oil; they employed profit and total heat duty or generated organic waste as two objectives. In addition, PBP, net present value or worth (NPV), and the internal rate of return can be used as one economic criterion for the MOO of chemical processes; details on these economic criteria can be found in [41,43].

**Energy Criteria:** The quantity of energy (in the form of steam, electricity, and/or fuel) is used to maximize the energy efficiency of process systems via MOO. Energy used in different forms can be combined into a single quantity using the efficiency of producing them. For example, Singh and Rangaiah [50] used efficiencies of 0.9 and 0.36, respectively, for steam and electricity production, and they justified these values. Even then, the quantity of energy, which is based on the first law of thermodynamics, does not explicitly consider the quality of energy based on the second law of thermodynamics, which can quantitatively measure the available and unavailable energy of the system and thus enhance the energy-based analysis. Hence, energy consumption and exergy loss (or efficiency) are two alternate criteria in the optimization of chemical processes. Details for calculating exergy and exergy efficiency can be found in the book by Seader and Henley [51]. Belfiore et al. [52] optimized the natural gas regasification process for minimizing investment and maximizing exergy efficiency simultaneously. Safari and Dincer [53] employed total exergy efficiency and methane production rate as the criteria to optimize an integrated wind power system for hydrogen and methane production.

The concept of exergy has been extended to exergoeconomic and exergoenviromental analysis. The former combines exergy and economic criteria into a single objective, while the latter combines exergy and environmental impact criteria into a single objective [54]. These two criteria can be directly used as two objectives or together with other criteria such as the production rate and recovery rate, to form an MOO problem to optimize processes. For example, Aghbashlo et al. [55]

employed exergoeconomic and exergoenviromental criteria as two objectives for the MOO design of an experimental setup for the continuous synthesis of solketal through glycerol ketalization with acetone.

**Environmental Criteria:** With the increasing concern on environmental pollution and global warming, sustainability and life cycle analysis (LCA) have become important for designing sustainable and environmentally friendly processes. Goedkoop et al. [56] proposed ECO99 for evaluating the sustainability and quantifying the environmental impact of the process, which is consistent with the philosophy of LCA and sustainability in the design of chemical processes. This methodology is based on evaluating three major damage categories: human health, ecosystem, and quality and resources depletion; and each category is divided into 11 sub-categories. ECO99 has been used as an objective in MOO studies. For example, Sánchez-Ramírez et al. [57] formulated a tri-objective optimization problem with ECO99, TAC, and individual risk as objectives for optimizing hybrid intensified downstream separation of biobutanol. The same three objectives have been employed to optimize reactive distillation processes for the eco-efficient production of ethyl levulinate [48]. Xu et al. [58] developed a vector-based multi-attribute decision-making method with weighted MOO for enhancing the sustainability of process systems.

The environmental impact of emissions reflects the efficiency of resource utilization. $CO_2$ emissions contribute significantly to global warming, and hence it is commonly employed as the criterion to quantify environmental impact and for the MOO of chemical processes. In this approach, the energy consumption including heating, cooling, and electricity duties is converted into equivalent fuel oil consumption to estimate $CO_2$ emission; more detailed calculations can be found in [59]. As mentioned above, exergoenviromental analysis does not measure the economic performance of the process, but it quantifies the impact of the designed process on environmental sustainability. Hence, it has become an important criterion used in MOO problems to design or retrofit a process [55].

One of the few studies using many objectives is by Sharma et al. [60], who analyzed the trade-off among the economic and environmental criteria of two recovery processes (one for solvent and another for volatile organic components) by solving nine MOO problems, each having two to 11 objectives. They also examined several aggregate environmental indicators, namely, the Potential Environmental Impact, IMPact Assessment of Chemical Toxics 2002+ (IMPACT), Green Degree, and Inherent Environmental Toxicity Hazard as well as their individual components as simultaneous objectives. They finally concluded that the optimization for individual categories is not necessary in the case of IMPACT, but it is required for other aggregate indicators to explore trade-offs among their individual components.

**Control Criteria:** Traditionally, chemical processes are first designed based on steady-state simulation and economic criteria followed by the synthesis of their control structures. Accordingly, the control system design begins from the designed process, which may lead to poor dynamic operability under process disturbances due to the trade-off between the optimal steady state and controllability of the process. Hence, this sequential approach may require iterations between process design and control system design for resolving conflicting objectives. Alternatively, process design and control can be conducted simultaneously to resolve the trade-off between design and control.

There are three main approaches for the simultaneous design and control of a process: (1) a controllability index-based approach, (2) a dynamic optimization-based approach, and (3) a robust control-based approach [61]. The first approach uses a controllability index to quantify the process closed-loop dynamic performance; controllability indexes such as relative gain array, condition number, and disturbance condition number are calculated using a steady-state model of the process. Then, one or more of these controllability indexes together with other steady-state criteria for economics are used to simultaneously design and control a given process, which can be solved using MOO [62,63]. The second approach utilizes dynamic optimization to obtain the optimal configuration of the process. For this, the dynamic performance under various disturbances are measured using a worst-case scenario, integral square error, integral absolute error, etc. Schweiger and Floudas [64] employed MOO in the design of a distillation process by using a dynamic optimization approach. The third approach is a

robust control-based approach, which estimates the bounds on process variables that determine the process flexibility, stability, and controllability of the system by using approximated robust models with uncertainty. The application of MOO via this approach is still open for researchers investigating the integration of design and control of chemical processes.

Model predictive control (MPC) is a widely used multivariable control strategy having better control performance than traditional proportional–integral control. However, tuning an MPC controller is challenging and important because of its dramatic impact on the control performance. Recently, Feng et al. [65] treated the tuning of an MPC controller as an MOO problem, where the two objectives are (i) the sum of squares of tracking errors between the output and reference trajectory and (ii) the sum of increments of manipulated variables. This tuning method has been used in the design of MPC for the operation of extractive dividing a wall column to obtain good control performance.

**Other Criteria:** Process safety is an important aspect of process synthesis and design. Among safety criteria, the Inherent Safety Index (ISI) developed by Heikkilä [66] ranks the inherent safety level of the chemical process based on the main and side reactions, parameters including pressure, temperature, yield, heat of reaction, inventory, flammability, toxicity, explosiveness, corrosion, equipment type, and process structure. Hassim et al. [67] developed the Inherent Occupational Health Index (IOHI) to evaluate the process in the initial stages of research and development based on the potential of working activities and process conditions that may harm workers. Later, Teh et al. (2019) [68] extended it to the Health Quotient Index (HQI) to quantify the health risk from fugitive emissions. Further, they utilized ISI and HQI together with the Potential Environmental Index evaluated by Waste Reduction (WAR) algorithm to formulate an MOO problem for optimizing the 1,4-butanediol production process. Eini et al. [69] proposed an MOO framework that employs Quantitative Risk Assessment (QRA) together with economic cost to optimize chemical processes.

A few studies have employed MOO in parameter estimation by using root mean squared error (RMSE), relative variance, and absolute error as evaluation criteria. For example, Punnapala et al. [70] employed MOO for parameter estimation in phase equilibrium; both the objectives are RMSE, one for activity coefficients, and another for heat of mixing. Bonilla-Petriciolet et al. [71] applied MOO for reconciling phase equilibrium data using RMSEs for temperature, pressure, and vapor/liquid mole fractions as separate objectives. Soares et al. [72] applied the RMSE of molar fraction, the mass of ethanol product, and the total mass collected from each batch to formulate a triple objective optimization to estimate the process and equipment parameters of an ethanol distillation process model based on their experimental data.

## 5. Computational Aspects

This section covers computational aspects of MOO, namely (1) techniques for solving MOO problems, (2) Pareto ranking methods for selecting one of the Pareto optimal solutions, and (3) available software for both (1) and (2). In addition, surrogate-assisted MOO is briefly covered in Section 5.1.5.

### 5.1. MOO Techniques

Recall that the selected applications reviewed earlier have mostly employed the weighted sum technique [73,74], ε-constraint technique [75], and some genetic algorithms (such as Nondominated Sorting Genetic Algorithm II (NSGA-II) [76]). In fact, there are many more MOO techniques, which can be classified in different ways. Broadly speaking, there are two approaches for solving MOO problems. One approach is to transform the MOO problem into one or a series of SOO problems, and then solve the resulting SOO problems by an established technique for SOO. Two common techniques following this approach are weighted sum and ε-constraint techniques. Here, this approach is referred to as the single-objective approach (SOA), where multiple objectives are adapted or transformed to a single objective by some strategy. The second approach is the modification of SOO techniques to solve problems with multiple objectives. It is referred to as multi-objective approach (MOA). Techniques

following this approach are metaheuristics (or stochastic algorithms) that are mostly population-based and include evolutionary algorithms.

In essence, the SOA modifies the problem (i.e., transforming an MOO problem into an SOO problem) to exploit the available SOO techniques whereas MOA adapts SOO techniques to solve the MOO problem directly. The transformation of multiple objectives (i.e., a vector) into one objective (i.e., a scalar) is known as scalarization; see Jahn [77] for a mathematical presentation of scalarization. The classification of MOO techniques into two groups, SOA and MOA, is intuitive and aptly captures the two alternate ways to develop techniques for solving MOO problems. The authors have not seen this way of classifying MOO techniques in the literature. In this section, the three popular techniques, namely, $\varepsilon$-constraint, weighted sum, and metaheuristics/stochastic optimization techniques as well as their merits and limitations are described before discussing the relative merits of SOA and MOA in Section 5.1.4.

The three popular techniques are described by reference to the following tri-objective optimization problem:

Minimize

$$f_1(x_1, x_2, \ldots, x_n) \tag{6}$$

Minimize

$$f_2(x_1, x_2, \ldots, x_n) \tag{7}$$

Minimize

$$f_3(x_1, x_2, \ldots, x_n) \tag{8}$$

with respect to:

$$x_1, x_2, \ldots, x_n$$

subject to:

$$x_i^L \leq x_i \leq x_i^U \quad \text{for} \quad i = 1, \tag{9}$$

$$h_j(x_1, x_2, \ldots, x_n) = 0 \quad \text{for} \quad j = 1, \tag{10}$$

$$g_j(x_1, x_2, \ldots, x_n) \leq 0 \quad \text{for} \quad j = m + 1, \tag{11}$$

The above problem has n decision variables ($x_1, x_2, \ldots, x_n$), which can be continuous or discrete, lower and upper bounds on them (Equation (9)), m equality constraints (Equations (10)), and (p–m) inequality constraints (Equation (11)). For simplicity, all three objectives (Equations (6)–(8)) are taken to be for minimization; maximization objectives, if present, can be changed to the minimization type or MOO techniques (some of which are described below) can be suitably modified. Inequality constraints (Equation (11)) are assumed to be more than or equal to type; if some inequality constraints are less than or equal type, then they can be changed to more than or equal to type.

### 5.1.1. $\varepsilon$-Constraint Technique

In this technique, any one of the objectives is retained as the (primary) objective to be optimized, whereas all the remaining objectives are appropriately included as inequality constraints. For example, in the $\varepsilon$-constraint technique, the tri-objective optimization problem in Equations (6)–(11) becomes the following SOO problem with only $f_3(x_1, x_2, \ldots, x_n)$ as the objective.

Minimize

$$f_3(x_1, x_2, \ldots, x_n) \tag{12}$$

With respect to:

$$x_1, x_2, \ldots, x_n \tag{13}$$

Subject to:

$$x_i^L \leq x_i \leq x_i^U \quad \text{for } i = 1, 2, \ldots, n \tag{14}$$

$$h_j(x_1, x_2, \ldots, x_n) = 0 \quad \text{for } j = 1, 2, \ldots, m \tag{15}$$

$$g_j(x_1, x_2, \ldots, x_n) \leq 0 \quad \text{for } j = m+1, m+2, \ldots, p \tag{16}$$

$$f_1(x_1, x_2, \ldots, x_n) \leq \varepsilon_1 \tag{17}$$

$$f_2(x_1, x_2, \ldots, x_n) \leq \varepsilon_2. \tag{18}$$

In the above SOO problem, two of the three objectives, namely, $f_1(x_1, x_2, \ldots, x_n)$ and $f_2(x_1, x_2, \ldots, x_n)$, are changed into two inequality constraints (Equations (17) and (18)) to keep them below some chosen values, namely, $\varepsilon_1$ and $\varepsilon_2$, respectively, while minimizing the (primary) objective $f_3(x_1, x_2, \ldots, x_n)$. Note that there is a different $\varepsilon$ for each objective that is changed into an inequality constraint. For completely defining the SOO problem in Equations (12)–(18), suitable values for $\varepsilon_1$ and $\varepsilon_2$ must be given. Then, the SOO problem can be solved using an appropriate SOO technique to find one of the Pareto optimal solutions. As in the weighted sum technique, one can choose different sets of values for $\varepsilon_1$ and $\varepsilon_2$, and then solve a series of SOO problems to find many Pareto optimal solutions.

Similar to the weighted sum technique (described below), the $\varepsilon$-constraint technique is intuitive, simple, requires values for $\varepsilon$s, and has been used in CPE since the 1970s. Both techniques require the solution of n SOO problems; each problem minimizes one objective while ignoring the others, but retaining all bounds and constraints in the MOO problem (Equations (6)–(11)) for normalization (in the weighted sum technique) or choosing values of $\varepsilon$s (in the $\varepsilon$-constraint technique). Both weighted sum and $\varepsilon$-constraint techniques give only one Pareto optimal solution and involve the solution of many SOO problems for finding many Pareto optimal solutions. Unlike the weighted sum technique where weights and the corresponding Pareto optimal solution found are not linearly related, values of $\varepsilon$s (i.e., $\varepsilon_1$ and $\varepsilon_2$ in tri-objective optimization) can be chosen in suitable increments to obtain evenly distributed Pareto optimal solutions. On the other hand, with the addition of inequality constraints for objectives other than the primary objective, the SOO problem arising from using the $\varepsilon$-constraint technique is more difficult to solve or may not have a solution.

### 5.1.2. Weighted Sum Technique

As the name implies, all objectives in a given MOO problem are combined into one single objective by using weighted sum. In this technique, the three objectives in Equations (6)–(8) can be combined into one objective given by Equation (19).

Minimize

$$w_1 f_1(x_1, x_2, \ldots, x_n) + w_2 f_2(x_1, x_2, \ldots, x_n) + w_3 f_3(x_1, x_2, \ldots, x_n). \tag{19}$$

Here, $w_1$, $w_2$ and $w_3$ are suitable weights for the three objectives. Except for this combined objective, the rest of the MOO problem (i.e., Equations (9)–(11)) remains the same. Weights are usually less than 1.0, and the sum of all weights is unity (i.e., $w_1 + w_2 + w_3 = 1.0$). Hence, values for at least n-1 (here, 3–1 = 2) weights have to be provided. With the weighted sum of objectives, the MOO problem in Equations (6)–(11) becomes an SOO problem (with Equation (19) as the objective, and Equations (9)–(11) as bounds and constraints. Any suitable SOO technique can be employed for solving this resulting SOO problem.

Thus, the weighted sum technique is simple and straightforward, and it uses established SOO techniques. However, it requires suitable values for weights. One strategy is to solicit inputs from decision makers (i.e., experts in the application area) on the relative importance of objectives, and then use the inputs obtained to determine suitable weights for the objectives. This strategy was employed in [25,27], which were reviewed earlier. The solution obtained by solving the resulting SOO problem will be only one of the Pareto optimal solutions.

Alternately, sets of values of weights can be chosen to cover the entire range for each weight (i.e., 0.0 to 1.0) and then solve the SOO problem for each set of weight values. In other words, this requires the solution of a series of SOO problems that differ only in the weights used in the weighted sum objective. The solution of the SOO problem with each set of weights will give one Pareto optimal solution. Obviously, there will be numerous sets of weight values, which will increase with the number of objectives. This strategy increases computational effort, may not provide uniformly distributed Pareto optimal solutions (because the relationship between weights and the Pareto optimal solution is non-linear), and does not find optimal solutions in the non-convex region of the Pareto optimal curve.

In addition, the objectives in an MOO problem such as Equations (6)–(11) may have very different magnitudes and/or significance. Both these can be accommodated by the normalization of each objective; e.g., the minimization objective, $f_1(x_1, x_2, \ldots, x_n)$ can be normalized by dividing by its lowest possible value obtained by solving an SOO problem to minimize $f_1(x_1, x_2, \ldots, x_n)$ while ignoring other objectives, but retaining all bounds and constraints in the MOO problem. This in turn will require the solution of n SOO problems prior to solving the MOO problem by the weighted sum technique.

Besides the weighted sum technique for the scalarization of multiple (a vector of) objectives, others include max–min and weighted product techniques. These are presented as a part of scalarization, value function, or aggregation methods in the literature. For details on alternates to the weighted sum technique, see the books by Deb [76] and Miettinen [78]. One popular approach for finding weights, which can be used in the weighted sum technique, is the analytical hierarchy process [79] that involves a pairwise comparison of objectives. This was employed in MOO applications studied by Ooi et al. [21] and Mano et al. [28], which are reviewed in Section 3.2.

### 5.1.3. Metaheuristics/Stochastic Optimization Techniques

Many metaheuristics or stochastic optimization techniques (since they use random numbers in the search for optimum) are available for solving SOO problems, and their number is continually increasing. Many of them have been adapted for solving MOO problems; popular metaheuristics adapted for MOO include evolutionary algorithms such as genetic algorithms [80] and differential evolution [81], and others such as particle swarm optimization [82] and ant colony optimization [83]. Some metaheuristics are not population-based (i.e., they use only one solution at a time); simulated annealing [84] is one such algorithm, and it was also adapted for solving MOO problems [85].

In general, evolutionary algorithms have been commonly used for MOO applications in CPE. Hence, the main steps are briefly described in the following. Evolutionary algorithms are inspired by the evolution of plants and animals in nature, employ a population of solutions (say, population size of N), and their iterations are termed generations. The population in the current generation and for the next generation are referred to as parent and child (offspring) population, respectively.

***Step 1—Initialization of the population***: An initial population of N individuals (or trial solutions or simply solutions) is generated, often randomly in the entire search space defined by lower and upper bounds. Values of all the objectives and constraints (both inequality and equality) are calculated at each of these N trial solutions. These N individuals form the parent population for the next step.

***Step 2—Generation of Child Population:*** This step of generating a child population varies from one evolutionary algorithm to another, and it has several sub-steps. In general, it consists of selecting a few individuals from the parent population, crossover among these selected individuals, and the mutation of resulting individuals. Thus, N new individuals are generated as the child population, and the values of all objectives and all constraints are calculated at each of these N new solutions.

***Step 3—Choosing Individuals for Next Generation:*** For this, parent and child populations are combined. Then, N individuals from this combined population are chosen for the next generation, considering feasibility (to ensure satisfaction of constraints), dominance (to achieve improvement in objective values), and crowding distance (for the wider spread) of solutions. One popular strategy for this selection is the non-dominant sorting based on the dominance concept (i.e., a solution is dominant over another if the former has objective values that are not inferior to those of the latter and has better

value for at least one objective). In case constraints are handled by the penalty function approach, then N individuals are chosen for the next generation based on dominance and crowding distance only. In short, the chosen N individuals are better than others in the combined population and form the parent population for the next generation.

*Step 4—Checking Stopping Criteria:* The most common stopping or termination criterion in metaheuristics is the maximum number of generations (iterations). Alternate criteria based on search progress (i.e., improvement in objective values) are possible. For example, two such criteria and their evaluation for the MOO of complex processes modeled by process simulators are presented by Rangaiah et al. [86]. If the specified stopping criteria are satisfied, iterations stop; else, iterations continue from Step 2.

Details including sample calculations of a number of evolutionary algorithms such as multi-objective genetic algorithm, NSGA-II, strength Pareto evolutionary algorithm and Pareto-archived evolution strategy are available in [76]. Metaheuristics including evolutionary algorithms for SOO problems differ from their respective adaptations for MOO problems, mainly in Step 3. In the case of an SOO problem, the population for the next generation is chosen considering only the values of the single objective (besides feasibility), whereas values of all objectives (through dominance of a solution) are considered in solving an MOO problem.

### 5.1.4. Single versus Multi-Objective Approach

Besides the weighted sum and the $\varepsilon$-constraint techniques, techniques such as the weighted metric technique, value function method (of which weighted sum technique is one special case), goal programming, and interactive methods belong to SOA. They are classical methods [76] and involve the scalarization of multiple objectives into one. Interactive methods present the Pareto optimal solutions found so far to the decision maker and seek her/his preferences for the subsequent search. They may require the decision maker's inputs or preferences, even in the beginning. Interactive methods also transform the given MOO problem into an SOO problem and then solve the SOO problem. This is repeated many times requiring the decision maker's preferences in between solving two slightly different SOO problems. The interactive methods and their applications in CPE can be found in [87].

The merits of the techniques in SOA are its simple concept and procedure, utilizing established SOO techniques and programs, and often computationally efficient for finding a few Pareto optimal solutions. However, the limitations of these techniques are additional inputs/preferences (such as weights and $\varepsilon$s) are required a priori (without any quantitative knowledge on trade-off among the objectives), the need to solve SOO problems many times, and only some Pareto optimal solutions are found. Metaheuristics may be appropriate for solving SOO problems that are multi-modal (i.e., having multiple optima), involve discontinuous objective and/or constraint functions, and require complex simulation (e.g., using a process simulator) for evaluating objectives and/or constraints. In such cases, it is better to use MOA to solve an MOO problem directly using a metaheuristic technique.

MOA to find Pareto optimal solutions followed by the selection of one of the Pareto optimal solutions is the ideal methodology for solving MOO problems, according to Deb [76]. Techniques in this approach are metaheuristics modified for MOO (mainly in Step 3 described above); they do not require any inputs from the decision maker, can handle discontinuous and highly nonlinear functions, can find many Pareto optimal solutions over a wide range, and are more likely to find the global Pareto optimal front. However, they are less effective for problems with equality constraints [88] and waste computational effort to find many Pareto optimal solutions, of which only one may be used.

Four out of the 14 studies on selected MOO applications reviewed in Section 3 found only one Pareto optimal solution. The remaining 10 studies found many Pareto optimal solutions using MOA (for 6 applications) or SOA (for 4 applications). Further, Kundu et al. [25] found that MOA using a genetic algorithm provides a uniform and wider Pareto optimal front compared to solving the equivalent SOO problem via the weighted sum technique. Many Pareto optimal solutions provide knowledge on the quantitative variation and trade-off among objectives, which can motivate further

technological developments. As noted earlier, Fuentes-Cortes et al. [19] observed that a wide range of trade-off solutions between cost and greenhouse gas emissions can stimulate the development of biogas markets.

Hence, it is recommended to find as many Pareto optimal solutions as possible for useful and perhaps unexpected results, even though more computational effort is required and only one of the optimal solutions may be implemented. SOA can be used for applications that involve equality constraints and continuous functions, whereas MOA (i.e., metaheuristics) is suggested for applications involving discontinuous and/or multi-modal functions or complex simulations.

### 5.1.5. Surrogate-Assisted Multi-Objective Optimization

The use of non-gradient evolutionary methods to solve MOO problems (i.e., swarm, genetic algorithm, and simulated annealing) can lead to large numbers of calculations of each objective function. Where one or more objective function calculations requires a simulation involving numerous equations or is time dependent, then the computational time required to solve the MOO problem is directly related to the number of objective function evaluations and the computational time for each evaluation. In these situations, the time required by the evolutionary algorithm itself is a minor proportion of the total calculation time. By replacing the full simulation with a surrogate model, the total calculation time for MOO can be reduced dramatically. Lambert et al. [89] provided a chemical process example solved by both the full Aspen Plus simulation and a surrogate approximation, which was tuned to the Aspen model. The MOO calculation time per generation (population of 50) reduced from 4800 s to just 0.4 s. Tuning the surrogate model prior to solving the MOO problem is the simplest approach. However, it requires full understanding of the solution space and development of the surrogate model a priori.

Often the solution space evolves with the optimization; therefore, it is necessary to embed the tuning of the surrogate model into the MOO procedure. The pioneer in this area is Ray et al. [90], who developed a surrogate-assisted framework and have since extended this framework to allow for multiple methods of tuning the surrogate models [91]. Diaz-Manriquez et al. [92] reviewed the different strategies for surrogate-assisted MOO. A good example of a computationally intensive problem in chemical engineering is pressure swing adsorption (PSA). The adsorption bed requires temperature-dependent cyclic boundary conditions. The cycle time and the bed geometry are important design variables, as are the pressure levels at each point in the cycle. Beck et al. [36] claim a 5-fold reduction in computational effort by using the surrogate modeling approach compared with a standard MOO approach.

### 5.2. Selection or Ranking Techniques

As noted above, it is desirable to find many Pareto optimal solutions before selecting one of them for implementation. Then, the question is how to choose one of the Pareto optimal solutions, which are equally good from the point of objectives in the MOO problem. Additional information and/or the relative importance of objectives (referred to as higher order information by Deb [76], who illustrates it with a simple example of car purchase) are required for selecting one of the Pareto optimal solutions. Such information is termed as decision maker's inputs or preferences in MCA literature that deals with the ranking of Pareto optimal solutions for selecting one of them. Many ranking techniques for Pareto optimal solutions have been proposed and discussed in MCA literature, but their use in CPE is very limited.

The ranking of Pareto optimal solutions obtained for CPE applications involves three steps: the normalization of objectives (which are likely to have significantly different magnitudes), the weighting of objectives, and finally the ranking of optimal solutions. There are several ways for the normalization of objectives, the number of weighting techniques, and many ranking techniques. Some weighting techniques use variation (such as the standard deviation or variance) of values of each objective, and they do not need inputs from the decision maker, whereas others require inputs from

the decision maker. The mean weighting technique (of assigning equal weights for all objectives) is the simplest. A brief description of four normalization techniques and 10 ranking techniques is available in [10], in which the authors analyzed the sensitivity of 10 ranking techniques to changes in objective values, through simulation. The recent paper by Hafezalkotob et al. [12] has an overview of weighting and ranking techniques as well as their use in the open literature.

Wang and Rangaiah [13] presented the principle and algorithm of 10 selection/ranking techniques, which were chosen considering simplicity, the ability to handle objective values with significantly different magnitude, and user inputs required. They also developed an MS Excel program (covered in the next sub-section) and used it for evaluation of the chosen 10 ranking techniques on many mathematical and CPE application problems. Based on the results, they recommended the technique for order of preference by similarity to ideal solution (TOPSIS), gray relational analysis (GRA), and simple additive weighting (SAW) for ranking Pareto optimal solutions. The basis of these methods is as follows. According to SAW, the score of each optimal solution is computed by summing the products of normalized objectives with respective weights, and the selected solution is that having the largest score. GRA employs a gray relational coefficient to quantify the similarity between each optimal solution (i.e., its objective values) and the ideal solution (i.e., having the best value of each objective). It is probably the only ranking technique that does not require weights or inputs from a decision maker. The chosen optimal solution by TOPSIS is that having the smallest Euclidean distance from the ideal solution and also the largest Euclidean distance from the negative-ideal solution (i.e., having the worst value of each objective).

*5.3. Software*

Many open-source and some commercial software are available for solving MOO problems using the techniques outlined above. For academic applications, most of them are freely available from different web sites; for example, Coello Coello [93] and Mittelmann [94] provide, on their personal websites, many open-source programs implemented in different software platforms for solving MOO problems. Table 4 lists MOO programs known to us and/or found by a quick search of the Internet and not by an exhaustive search, partly because new/updated programs can be expected in the coming years. Here, the purpose of Table 4 is to provide a ready reference to some available programs for readers interested in solving their MOO problems. Most of these programs are implemented in Matlab and Python.

In Table 4, jMetal [95] is an object-oriented java-based MOO library consisting of a number of classical and modern multi-objective evolutionary algorithms (MOEAs), which can be configured and executed via jMetal's Graphical User Interface (GUI). The MOEA framework is another open-source java-based library for developing and experimenting with SOO and MOO algorithms [96].

**Table 4.** Some available software for solving MOO problems.

| Name | Brief Description | Reference |
| --- | --- | --- |
| ParadisEO-MOEO | A C++-based open source objective-oriented framework, providing visualization facilities and on-line definition of parameters. | [97] |
| jMetal | An object-oriented java-based MOO library framework. | [95] |
| EMOO | MS Excel based NSGA-II program for non-linear constrained MOO. | [98] |
| MOSQP | Sequential Quadratic Programming (SQP) method in Matlab for differentiable constrained MOO. | [99] |

**Table 4.** *Cont.*

| Name | Brief Description | Reference |
|---|---|---|
| OTL | A C++ library for solving MOO problems, and it is then extended to Python modules in PyOTL software. | [100] |
| PyGMO | Parallel global multi-objective optimizer coded in Python for non-linear constrained MOO problems. | [101] |
| IMODE | Integrated Multi-Objective Differential Evolution program in MS Excel for non-linear constrained MOO. | [5] |
| MOEA Framework | A Java library for developing and experimenting with single and MOO algorithms. | [96] |
| PlatEMO | A powerful Matlab-based software for solving MOO problems, and it includes more than 50 multi-objective evolutionary algorithms. | [102] |
| TSEMO | Thompson sampling efficient MOO algorithm coded in Matlab for constrained non-linear MOO problems. | [103] |
| MOGOA | Multi-Objective Grasshopper Optimization program in Matlab for constrained non-linear MOO problems. | [104] |
| NAGA-III | Implementation of NSGA-III algorithm in Matlab. | [105] |
| PISA | For solving constrained MOO problems in Matlab, C, and java. | [106] |
| DAKOTA | A C++-based toolkit for solving MOO and single-objective optimization problems. | [107] |
| NGPM | Reference-point based NSGA-II Program in Matlab for solving MOO problems. | [108] |
| MO-HERPA | A commercial software for solving large engineering problems. | [109] |

Matlab provides the "*gamultiobj*" function in its global optimization toolbox [110]; it employs the NAGA-II algorithm for solving MOO problems and includes GUI to visualize the progress of the optimization via real time plots. Lin [108] presented a powerful MOO solver, which was named NGPM (NSGA-II Program in Matlab), on Matlab file exchange community; it employs reference-point based NSGA-II and has GUI as well. PlatEMO [102] is an extensive software based on the Matlab platform; it has more than 50 multi-objective evolutionary algorithms (MOEAs) and more than 100 multi-objective test problems. It also provides GUI, enabling users to easily compare several algorithms at a time and collect statistical results in MS Excel files. Note that this software is completely open source, and its source code is currently available at http://bimk.ahu.edu.cn/index.php?s=/Index/Software/index.html.

In recent years, Python is becoming increasingly popular in academic research and education since it is fast and completely open source, leading to the development of more and more open-source programs in Python. PyGMO [101] is a powerful scientific library for massively parallel optimization, which can be used to solve constrained, unconstrained, single-objective, multi-objective, continuous, and integer optimization problems as well as stochastic and deterministic problems. PyGMO is also implemented in C++; it is named PaGMO.

Although many open-source programs implemented in Matlab, Python, C++, and Java are available, relevant programming experience is required to use them, which limits their application. On the other hand, MS Excel is readily available and used by many researchers, engineers, and students. Realizing this, Sharma et al. [98] developed an MS Excel-based MOO (EMOO) program, which employs binary-coded NSGA-II and the maximum number of generations (MNG) as the termination criterion. This program is the first of its type for MOO in MS Excel. In this program, MS Excel worksheets are used to define the MOO problem and specify algorithm parameters for the calculation of objective functions and constraints and to display results. Visual Basic for Application (VBA) was used for implementing

binary-coded NSGA-II. Wong et al. [111] expanded the capabilities of the EMOO program by including real-coded NSGA-II and two progress-based termination criteria (besides MNG). Rangaiah et al. [86] evaluated these two termination criteria in EMOO and I-MODE programs (Table 4) for the MOO of complex chemical processes simulated by commercial simulators.

The final step of MOO is to select one of the obtained non-dominated (Pareto optimal) solutions as the optimal solution for implementation. For this selection, Wang and Rangaiah [13] developed an MS-Excel based program using Visual Basic for Application (VBA); this program includes 10 common selection/ranking algorithms. This is probably the only program in MS Excel, and it has many selection/ranking algorithms.

## 6. Discussion

As seen in the previous sections, MOO with more than one objective is more appropriate in view of conflicting objectives in many applications. Figures 3 and 5 clearly indicate that the role of MOO in the CPE field continues to increase. MOO applications in CPE cover a wide range such as for designing a new process, for retrofitting/revamping an existing process, for process control, for planning and scheduling, for supply chain optimization, for parameter estimation, and for product design. Therefore, process optimization in the future is expected to be mainly for multiple objectives. Tables 5 and 6 summarize the similarities and differences between SOO and MOO, respectively.

The previous sections covered many aspects of MOO such as steps in MOO, MOO applications in CPE, diverse objectives in process applications, MOO techniques, surrogate-assisted MOO, ranking techniques for selecting one of the Pareto optimal solutions, and available programs for MOO. However, there are other topics not covered in this paper; these include the graphical representation of Pareto optimal solutions in the objective space (particularly, when there are more than three objectives), performance metrics for assessing the accuracy and quality of the Pareto optimal front obtained, and benchmark/mathematical problems for testing MOO algorithms/programs. Readers interested in these are referred to the comprehensive and detailed book by Deb [76].

Although MOO has become common in the CPE field, the following types of applications or topics received much less attention. MOO use for parameter estimation (i.e., for estimating unknown parameters in the model via fitting experimental data) and in process control is limited so far. The number of objectives in reported CPE applications is often two or three; on the other hand, progress is being made to solve MOO problems with many objectives efficiently. Hence, it is feasible to optimize CPE applications for many objectives. One important aspect that did not receive much attention in CPE is the selection of one of the Pareto optimal solutions, although there are many techniques and studies on this in MCA literature.

As described in Section 5, computer programs are readily available for MOO. Some of these programs (including MS Excel-based programs) are free, and some others are in software platforms such as Matlab. However, MOO programs based on MOA are not yet available in the popular process simulators such as Aspen Plus and Hysys. Hence, for applications involving process simulations in these simulators, users may have to resort to SOA (e.g., use weighted sum or $\varepsilon$-constraint techniques in conjunction with the available SOO program) or interface the simulator with the optimizer in another platform.

Industries may be using suitable MCA or ranking techniques for decision making such as selecting a vendor from potential vendors, for selecting one of the promising sites for a new plant, and for selecting a technology from several alternatives. However, the formulation and solution of optimization problems for multiple objectives in industrial practice are not clear. One reason for this is that industries usually do not communicate their ongoing research and development in the open literature. One commercial software (namely, HEEDS$^®$ [109]) has many optimization algorithms, including MOO algorithms. This software availability and this paper should motivate and lead to the beneficial utilization of MOO in industrial practice.

**Table 5.** Similarities between SOO and MOO.

| Feature | Similarity |
|---|---|
| *Decision Variables* | Continuous and/or integer variables with or without bounds on them. |
| *Constraints* | The problem may or may not have equality and/or inequality constraints. |
| *Type of Equations* | Objective(s) and/or constraints can be linear, non-linear, differential, and/or integral equations. |
| *Solution Techniques* | Both deterministic and stochastic (metaheuristics) optimization techniques can be used. |
| *Optimal Solutions* | An SOO problem can have unique or multiple optimal solutions (such as local and global optima). Similarly, an MOO problem can have local and global Pareto optimal fronts (with each front having multiple optimal solutions). |

**Table 6.** Differences between single-objective optimization (SOO) and MOO.

| Feature | SOO | MOO |
|---|---|---|
| *Number of Objectives* | Only one objective to minimize or maximize. | Two or more objectives, which can be the minimization and/or maximization type. |
| *Number of optimal solutions* | Usually only one optimal solution. | Many optimal solutions, which are known as Pareto optimal or non-dominated solutions. One single optimal solution only when all objectives are not conflicting. |
| *Multi-dimensional Spaces* | Only one multi-dimensional space for decision variables. | Two multi-dimensional spaces: one for objectives and another for decision variables. |
| *Development of Techniques* | Optimization techniques were originally developed for SOO problems with or without constraints. | There are two approaches for solving an MOO problem. One approach is to convert an MOO problem into an SOO problem for solution by an SOO technique. Another approach is to modify stochastic optimization techniques (metaheuristics) to handle multiple objectives for solving an MOO problem. |
| *Computational Time for Finding Optimal Solutions* | Deterministic techniques are faster compared to stochastic techniques, but the latter are more likely to find the global solution. | Deterministic techniques are faster if only a few Pareto optimal solutions are required. Stochastic techniques may be faster for finding many Pareto optimal solutions. In general, computational time is expected to increase with the number of objectives. |
| *Knowledge on Optimal Solutions* | Usually, limited to one optimal solution, either local or global optimum. | Quantitative variation and trade-off of objectives from the many optimal solutions in the Pareto optimal front, which can be a local or global optimal front. |
| *Selection of a Solution for Implementation* | It is straightforward, since only one or a few optimal solutions are found. Global optimum is often preferred. | Additional preferences and techniques are required for selecting one of the non-dominated (optimal) solutions, which are equally good from the perspective of objectives in the MOO problem. Owing to many choices for normalization, weighting, and ranking, the selection of one of the non-dominated solutions is itself another optimization problem. |

In summary, further research and development in the following directions are suggested.

- Novel applications of MOO in CPE such as for parameter estimation and control
- MOO of CPE applications with many objectives (and not just two or three objectives) and comprehensive analysis of the optimal results obtained
- Surrogate-assisted MOO for computationally intensive applications in CPE
- Efficient and reliable codes including properly parallelized codes for MOO
- Implementation of MOO codes in process simulators
- Effective techniques for constraints, particularly, equality constraints in stochastic optimization techniques (metaheuristics) for MOO
- Application and analysis of techniques for selecting one optimal solution from the Pareto optimal front obtained for CPE applications

**Author Contributions:** Conceptualization, G.P.R.; Investigation, G.P.R., Z.F., A.F.H.; Writing—Original Draft Preparation, G.P.R., Z.F., A.F.H.; Writing—Review and Editing, G.P.R., A.F.H.; Supervision, G.P.R. All authors have read and agreed to the published version of the manuscript.

**Funding:** This research received no external funding.

**Conflicts of Interest:** The authors declare no conflict of interest.

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
