# Peer review of "Multi-Objective Optimization Applications in Chemical Process Engineering: Tutorial and Review"

_processes, doi:10.3390/pr8050508_

Round 1

Reviewer 1 Report

This is an interesting hybrid review/tutorial on vector optimization in process engineering. The work is of good quality, and will be a good guide to researchers beginning work in this field. Only minor revisions are needed, as follows:

  1. In a review paper I consider it best practice to trace key ideas to their ultimate source (e.g., Kennedy and Eberhart for PSO, Haimes and Hall for epsilon constraint method, etc.). This point is not necessarily a strict rule, as it is not needed for "textbook information," but for newer ideas it is also useful for the reader seeking more details.
  2. The epsilon constraint method should be discussed before all other methods as it is generally considered as the most fundamental approach to vector optimization.
  3. The weighted sum method is discussed as a means of converting a vector optimization problem into an equivalent single-objective one. What about other aggregation methods? For example, max-min aggregation via fuzzy mathematical programming has interesting mathematical properties that are useful in optimization. Other aggregation approaches have also been reported - weighted product, OWA, etc.

Author Response

We thank the two reviewers for their valuable comments and suggestions. We have carefully considered and addressed them with additions, shown in blue color in the revised manuscript. Further, we made minor corrections for enhancing the presentation. Each of the reviewers’ comments (reproduced in italics) and our responses/actions taken are described below.

Reviewer 1

 Reviewer’s Comment: This is an interesting hybrid review/tutorial on vector optimization in process engineering. The work is of good quality, and will be a good guide to researchers beginning work in this field. Only minor revisions are needed, as follows:

Authors’ Response: We thank the reviewer for the compliments on our work and positive recommendation.

Reviewer’s Comment 1: In a review paper I consider it best practice to trace key ideas to their ultimate source (e.g., Kennedy and Eberhart for PSO, Haimes and Hall for epsilon constraint method, etc.). This point is not necessarily a strict rule, as it is not needed for "textbook information," but for newer ideas it is also useful for the reader seeking more details.

Authors’ Response: About 10 new references are added to provide ultimate source for the methods stated; these additions are shown in blue text in the manuscript.

Reviewer’s Comment 2: The epsilon constraint method should be discussed before all other methods as it is generally considered as the most fundamental approach to vector optimization.

Authors’ Response: As per this comment, epsilon constraint method is presented before weighted sum technique.

Reviewer’s Comment 3: The weighted sum method is discussed as a means of converting a vector optimization problem into an equivalent single-objective one. What about other aggregation methods? For example, max-min aggregation via fuzzy mathematical programming has interesting mathematical properties that are useful in optimization. Other aggregation approaches have also been reported - weighted product, OWA, etc.

Authors’ Response: Agree that there are other aggregation/scalarization methods. These are stated along with a few references, in a new paragraph at the end of Section 5.1.2 in the revised manuscript.

Reviewer 2 Report

Rangaiah et al. report about “Multi-Objective Optimization Applications in Chemical Process Engineering: Tutorial and Review”. The authors give a good overview of so far developed methods for single- and multi-objective optimization in the field of chemical engineering. Different approaches of either fundamental, empirical or statistical analysis are reviewed. In conjunction with short summaries of published work and applied software packages (commercial and open source), the reader receives with this review a useful tool to develop an optimization scheme for his own purpose. The content is of high quality as well as the grammar and spelling. Hence the manuscript is recommended for publication in Processes without revision, except some typos, for example:

wordings should be unified:  CO2 vs. CO2, ε constraint vs. ε-constraint

line 466: application

line 554: point at the end of the sentence is missing

line 561: “The word criteria are is used…”

line 543: “…can be found…”

There might be more mistakes. Please, check carefully before publication.

Author Response

We thank the two reviewers for their valuable comments and suggestions. We have carefully considered and addressed them with additions, shown in blue color in the revised manuscript. Further, we made minor corrections for enhancing the presentation. Each of the reviewers’ comments (reproduced in italics) and our responses/actions taken are described below.

Reviewer 2

Reviewer’s Comment: Rangaiah et al. report about “Multi-Objective Optimization Applications in Chemical Process Engineering: Tutorial and Review”. The authors give a good overview of so far developed methods for single- and multi-objective optimization in the field of chemical engineering. Different approaches of either fundamental, empirical or statistical analysis are reviewed. In conjunction with short summaries of published work and applied software packages (commercial and open source), the reader receives with this review a useful tool to develop an optimization scheme for his own purpose. The content is of high quality as well as the grammar and spelling. Hence the manuscript is recommended for publication in Processes without revision, except some typos, for example:

Authors’ Response: We thank the reviewer for the summary, compliments and recommendation on the publication of the manuscript.

Reviewer’s Comment 1: wordings should be unified:  CO2 vs. CO2, ε constraint vs. ε-constraint

line 466: application

line 554: point at the end of the sentence is missing

line 561: “The word criteria are is used…”

line 543: “…can be found…”

There might be more mistakes. Please, check carefully before publication.

Authors’ Response: We made corrections for all the above and also carefully checked and updated the entire manuscript for other minor typographical errors.